# The Effect of Extraction Methods on Preliminary Structural Properties and Antioxidant Activities of Polysaccharides from *Lactarius vividus*

**Zhou Xu** [1,†], **Shiling Feng** [1,†], **Jipeng Qu** [1,2], **Ming Yuan** [1], **Ruiwu Yang** [1] , **Lijun Zhou** [1], **Tao Chen** [1] **and Chunbang Ding** [1,*]

1  College of Life Sciences, Sichuan Agricultural University, Yaan 625014, China
2  College of Agricultural Science, Xichang University, Xichang 615000, China
*  Correspondence: dcb@sicau.edu.cn; Tel.: +86-083-562-5014
†  These authors contributed equally to this paper.

**Abstract:** Four polysaccharides (LVP-u, LVP-m, LVP-e, and LVP-h) were extracted from the fruiting bodies of *Lactarius vividus* by an ultrasonic-assisted extraction method, microwave-assisted extraction method, enzyme-assisted extraction method, and hot water extraction method, respectively. Then, the effect of extraction methods on yields, preliminary structural properties, and antioxidant activities was systematically investigated using the weighing method, chemical composition analysis, high-performance gel permeation chromatography (HPGPC), high-performance liquid chromatography (HPLC), Fourier-transform infrared spectroscopy (FT-IR), scanning electron microscope (SEM), radical scavenging, and metal ion chelating assays. Results demonstrated that the four *L. vividus* polysaccharides (LVPs) were all combined with protein, and the yield of LVP-u was higher than others. Molecular weight distribution, monosaccharide and amino acid compositions, and microstructures among the four LVPs were significantly different. Moreover, the LVPs showed significant antioxidant activities in a dose-dependent manner, and LVP-e demonstrated better antioxidant activities in superoxide anion radical scavenging activity assays and metal ion chelating activity assays, while LVP-u showed higher activity in its hydroxyl radical scavenging ability.

**Keywords:** *Lactarius vividus*; polysaccharides; extraction methods; preliminary structure properties; antioxidant activities

## 1. Introduction

Up to now, at least 12,000 species of mushrooms have been found, and about 2000 species among those are reported as edible [1]. Edible mushrooms are the most precious gifts of nature to human beings, which not only provide delicacies, but also serve as indispensable raw materials for pharmaceuticals, nutraceuticals, and cosmeceuticals [2]. Generally, mushrooms are considered to be valuable health food, rich in easily digestible proteins, dietary fibers, and minerals, and low in fat. Moreover, mushrooms also have been confirmed to contain various bioactive substances, including vitamins, phenolic acids, terpenoids, and sterols [3,4]. Moreover, in the 1960s, biologists found that polysaccharides from *Lentinula edodes* had strong antitumor activity against sarcoma 180 when implanted subcutaneously in mice. Mushroom polysaccharides gradually attracted the interests of many researchers [5], and subsequent studies also showed that polysaccharides and their derivatives purified from mushrooms possess antioxidant, immunomodulatory, antitumor, anti-gastric ulcer, antihyperglycaemic activities, etc. [6–9].

Recently, various extraction techniques, such as hot water extraction, alkali extraction, ultrasonic-assisted extraction, microwave-assisted extraction, and enzyme-assisted extraction have

been employed for the extraction of polysaccharides. However, among these techniques, a number of researchers make a choice according to the yield sometimes, not based on the comprehensive evaluation system related to structural characteristics and biological activities of polysaccharides as well. Song et al. [10] reported that *Bryopsis plumosa* polysaccharides extracted by sulfuric acid solution had the lowest molecular weight due to acid hydrolysis occurred in extraction process, but polysaccharide extraction with sodium hydroxide had the highest sulfate content and strongest antioxidant power. Jia et al. [11] compared the effects of different enzyme-assisted extraction methods on the bioactivity of *Agaricus blazei* polysaccharides, and compound enzyme extraction exhibited the strongest reducing power and highest scavenging rates on radicals. Yan et al. [12] confirmed that polysaccharides from *Amomum villosum* extracted by ultrasonic-assisted extraction contained the highest content of uronic acid and sulfate, and the best antioxidant activity. Moreover, some studies have revealed that different extraction methods have various effects on the microstructure of polysaccharides [13,14].

*Lactarius vividus*, a basidiomycetes belongs to *Lactarius*, is one of the most popular wild edible mushrooms distributed in China [15]. However, up to now, little attention has been paid to evaluate its preliminary structural properties and the antioxidant activities of polysaccharides from *L. vividus* (LVPs). Thus, in the present study, LVPs were extracted via the ultrasonic-assisted extraction method, microwave-assisted extraction method, enzyme-assisted extraction method, and hot water extraction method. Additionally, preliminary structural characteristics are analyzed by high-performance gel permeation chromatography (HPGPC), high-performance liquid chromatography (HPLC), Fourier-transform infrared spectroscopy (FT-IR), and scanning electron microscopy (SEM). Finally, antioxidant activities of four LVPs are investigated by free radical scavenging assays and metal ion chelating assays.

## 2. Materials and Methods

### 2.1. Materials and Reagents

Fruiting bodies of *L. vividus* were collected in the pine forests from Dazhou (Sichuan Province, southwestern China) in October 2016. They were authenticated in the Dazhou Institute of Agricultural Sciences based on their microscopic and macroscopic characteristics. The fruiting bodies were air-dried in an oven for 48 h at 50 °C, and afterwards were ground to ultrafine powder. Experimental material was extracted with petroleum ether, acetone, and ethanol by a Soxhlet extraction system to remove small molecular weight impurity.

Mannose, rhamnose, glucose, galactose, arabinose, glucuronic acid, galacturonic acid, fucose, asparaginic acid, glutamic acid, asparagine, serine, glutamine, histidine, glycine, threonine, citrulline, arginine, alanine, tyrosine, cysteine, valine, methionine, tryptophan, phenylalanine, isoleucine, leucine, lysine, hydroxyproline, proline, nitroblue tetrazolium (NBT), *o*-phthalaldehyde (OPA), dihydronicotinamide adenine dinucleotide (NADH), 9-Fluorenylmethyl chloroformate (FMOC), phenyl-3-methyl-5-pyrazolone (PMP), phenazine methosulfate (PMS), and ferrozine were purchased from Sigma Chemical Co. (St. Louis, MO, United States). The molecular weight standard substances of Dextran were provided by the National Standard Material Research Center. Other analytic reagents were obtained from Chengdu Kelong Chemical Factory (Chengdu, Sichuan, China).

### 2.2. Extraction of LVPs by Various Methods

#### 2.2.1. Ultrasonic-Assisted Extraction

The ultrasonic-assisted extraction process was based on the method of Tian et al. [16] with slight modifications. Briefly, 15 g of a pretreated sample was extracted with 450 mL distilled water at 70 °C for 1 h, utilizing an ultrasound-assisted extraction machine (KQ-300DV, Kunshan ultrasonic instrument co., LTD, Kunshan, Jiangsu, China) with ultrasound power setting at 240 W. The supernatant was collected, concentrated, and precipitated with four times volume ethanol for 12 h at 4 °C. Then, the precipitate

was isolated and dried to obtain crude polysaccharides. Finally, crude polysaccharide aqueous solution was subjected to the Sevag reagent (chloroform:butyl alcohol, 4:1, *v/v*) to remove free proteins and the dialysis bag (exclusion limit 8–14 kDa) to dislodge small molecular weight impurity. The non-dialyzable portion was freeze-dried to afford LVP-u.

### 2.2.2. Microwave-Assisted Extraction

The microwave-assisted extraction process was based on the method of Zhang et al. [17] with slight modifications. Briefly, 15 g of a pretreated sample was extracted with 450 mL distilled water at 75 °C for 30 min, utilizing a microwave synthesis reaction apparatus (UWave-1000, Shanghai Xinyi Microwave Chemical Technology Co. LTD, Shanghai, China) with microwave power at 400 W. The supernatant was collected, and the following procedures were consistent with Section 2.2.1. Finally, the sample was named LVP-m.

### 2.2.3. Enzyme-Assisted Extraction

The enzyme-assisted extraction process was based on the method of Zhu et al. [18] with slight modifications. Briefly, a 15 g pretreated sample was extracted with 450 mL enzyme solution (pH 5, cellulose:pectinase:trypsin ratio of 2:2:1, *v/v/v*) at 50 °C for 30 min. The supernatant was collected, and the following procedures were consistent with Section 2.2.1. Finally, the sample was named LVP-e.

### 2.2.4. Hot Water Extraction

The hot water extraction process was based on the method of Hou et al. [19] with slight modifications. Briefly, a 15 g pretreated sample was extracted with 450 mL distilled water at 80 °C for 2 h. The supernatant was collected, and the following procedures were consistent with Section 2.2.1. Finally, the sample was named LVP-h.

### 2.3. Yield Calculation of LVPs

The yield of polysaccharides (%) was calculated as the following formula:

$$\text{Extraction yield } (\%) = \frac{a}{b} \times 100 \tag{1}$$

where *a* is the weight of the LVPs and *b* is the weight of the *L. vividus* power.

### 2.4. Preliminary Structural Properties of LVPs

The total carbohydrate content of LVPs was determined by the phenol-sulfuric acid method [20], using d-glucose as standard. The protein content of LVPs was measured by the bicinchoninic acid/$CuSO_4$ (BCA) method [21], using BSA (bovine serum albumin) as the standard. The molecular weight distribution of the LVPs was evaluated by HPGPC combined with a refractive index detector (RID) (Agilent Technologies, Palo, Alto, CA, United States). The chromatographic conditions were as follows: injection volume: 20 μL, mobile phase: NaCl solution (0.1 M), flow rate: 0.5 mL/min, column temperature: 30 °C, detector temperature: 40 °C, and chromatographic column: PL aquagel–OH (7.5 × 300 μm). A standard curve was established using a series of dextran standard substances (Mw = 4320; 12,600; 73,800; 110,000; and 289,000).

The monosaccharide composition of LVPs was analyzed by a pre-column derivation HPLC method [22]. Briefly, 2 mL of polysaccharides solution (4 mg/mL) was mixed with 1 mL trifluoroacetic acid (TFA) in a sealed ampoule bottle and hydrolyzed at 100 °C for 6 h. After hydrolysis, residual TFA was removed with methanol under negative pressure. Then, the hydrolysate was dissolved to 3 mL with distilled water. Finally, 0.2 mL of solution was derivatized by 250 μL PMP (0.5 M) under alkaline conditions. The PMP-labeled monosaccharide sample was analyzed by a Agilent 1260 HPLC system equipped with a Zorbax Eclipse Plus C18 column (150 × 4.6 mm, 5.0 μm) and a diode array detector

(DAD) detector. The chromatographic conditions were as follows: injection volume: 10 μL; mobile phase: 0.05 M $KH_2PO_4$ (pH 6.9) with 15% (solvent A) and 40% (solvent B) acetonitrile, with the gradient 0%–8%–25% solvent B by a linear increase from 0–10–32 min; mobile phase flow rate: 1.0 mL/min; column temperature: 25 °C; detection wavelength: 250 nm.

The amino acid composition of the LVPs' protein regions was analyzed by the HPLC method. Briefly, 20 mg polysaccharides were hydrolyzed by 6 M HCl at 110 °C for 24 h. After hydrolysis, residual HCl was removed with methanol under negative pressure. Then, the hydrolysate was dissolved to 3 mL with 0.1 M HCl. Finally, amino acids were analyzed by HPLC equipped with a Zorbax Eclipse AAA column (150 × 4.6 mm, 5.0 μm) and a fluorescence detector (FLD) by online, pre-column derivatization, high-performance liquid chromatography that has been reported in detail by Sun et al. [23].

FT-IR spectra of LVPs were recorded by an IR spectrophotometer (FT-IR, Thermo Nicolet-380) in the range of 400–4000 $cm^{-1}$ using the KBr pellet method. The morphological characteristics of LVPs were observed by scanning electron microscopy (SU8220, Hitachi, Japan), after a freeze-dried polysaccharides sample was coated with gold powder in a vacuum and a scanning electron microscope image was magnified 10,000×.

### 2.5. Radical Scavenging Assays of LVPs

Polysaccharide samples and a positive control (Vc) were prepared to a series of concentration solutions (0.25, 0.5, 1.0, 2.0, 3.0, 4.0, and 5.0 mg/mL) using deionized water. A superoxide anion radical scavenging assay and hydroxyl radical scavenging assay were conducted to evaluate the radical scavenging abilities of LVPs.

The superoxide anion radical scavenging assay was carried out according to the method of Wang et al. [24], with slight modifications. Briefly, 50 μL of polysaccharides solution, 50 μL of NBT solution (156 μM), 50 μL of NADH solution (156 μM), and 50 μL of PMS solution (60 μM) were successively loaded to a sample cell of a 96-well plate. After the reaction mixture incubated at room temperature for 5 min, the absorbance at 560 nm was measured. The scavenging activity of the test sample on superoxide anion radicals was calculated by the following equation:

$$\text{Superoxide anion radical scavenging activity } (\%) = \left[1 - \left(\frac{Abs_1 - Abs_2}{Abs_0}\right)\right] \times 100 \tag{2}$$

where $Abs_0$ is the absorbance of deionized water reacted with the superoxide anion radical generating system, $Abs_1$ is the absorbance of the polysaccharide solution reacted with the superoxide anion radical generating system, and $Abs_2$ is the absorbance of the test sample solution only.

The hydroxyl radical scavenging assay was carried out according to the method of Giese et al. [25] with slight modifications. Briefly, 50 μL of the polysaccharide solution, 50 μL of the sodium salicylate solution (9 mM), 50 μL of the $FeSO_4$ solution (9 mM), and 50 μL of the $H_2O_2$ (0.025%, *w/v*) were successively loaded to a sample cell of a 96-well plate. After the reaction mixture incubated at 37 °C for 30 min, the absorbance at 562 nm was measured. The scavenging activity of the test sample on hydroxyl radicals was calculated by the following equation:

$$\text{Hydroxyl radical scavenging activity } (\%) = \left[1 - \left(\frac{Abs_1 - Abs_2}{Abs_0}\right)\right] \times 100 \tag{3}$$

where $Abs_0$ is the absorbance of deionized water reacted with the hydroxyl radical generating system, $Abs_1$ is the absorbance of polysaccharide solution reacted with the hydroxyl radical generating system, and $Abs_2$ is the absorbance of the test sample solution only.

*2.6. Metal Ion Chelating Assays of LVPs*

　　Ferrous ion and cupric ion chelating assays were conducted to evaluate the metal ion chelating capacities of LVPs, and EDTA-2Na was utilized for a positive control.

　　The ferrous ion chelating assay was carried out according to the method of Kalın et al. [26], with slight modifications. Briefly, 50 µL of polysaccharide solution, 100 µL of $FeSO_4$ solution (0.125 mM), and 50 µL of ferrozine solution (1.0 mM) were loaded to a sample cell of a 96-well plate successively. After the reaction mixture was incubated at room temperature for 5 min, the absorbance at 560 nm was measured. The $Fe^{2+}$ chelating activity was calculated by the following equation:

$$Fe^{2+} \text{ chelating activity } (\%) = \left[1 - \left(\frac{Abs_1 - Abs_2}{Abs_0}\right)\right] \times 100 \tag{4}$$

where $Abs_0$ is the absorbance of deionized water reacted with the $FeSO_4$–ferrozine reaction system, $Abs_1$ is the absorbance of the polysaccharide solution reacted with the $FeSO_4$–ferrozine reaction system, and $Abs_2$ is the absorbance of the sample only (deionized water instead of the $FeSO_4$–ferrozine reaction system).

　　The cupric ion chelating assay was carried out according to the method of Xu et al. [22], with slight modifications. Briefly, 50 µL of polysaccharide solution, 100 µL of $CuSO_4$ (0.1 mg/mL in 50 mM acetate buffer, pH 6.0), and 50 µL of pyrocatechol violet solution (4 mM) were loaded to a sample cell. Then, the mixture was incubated at room temperature for 20 min, and the absorbance at 632 nm was measured. The $Cu^{2+}$ chelating activity was calculated by the following equation:

$$Cu^{2+} \text{ chelating activity } (\%) = \left[1 - \left(\frac{Abs_1 - Abs_2}{Abs_0}\right)\right] \times 100 \tag{5}$$

where $Abs_0$ is the absorbance of deionized water reacted with the $CuSO_4$–pyrocatechol violet reaction system, $Abs_1$ is the absorbance of polysaccharide solution reacted with the $CuSO_4$–pyrocatechol violet reaction system, and $Abs_2$ is the absorbance of the sample only (deionized water instead of the $CuSO_4$–pyrocatechol violet reaction system).

*2.7. Statistical Analysis*

　　All the experiments were conducted in triplicate, and the data were shown in means ± SD. Statistical analysis was performed using analysis of variance (ANOVA) with the IBM SPSS Statistics 22 software, and differences were considered significant at $p < 0.05$.

## 3. Results

*3.1. Yields and Preliminary Structure Properties of LVPs*

　　The yields and chemical compositions of the LVPs are shown in Table 1. Based on weighing method, the yields of LVP-u, LVP-m, LVP-e, and LVP-h were 3.74%, 3.09%, 3.33%, and 3.63%, respectively. Moreover, LVP-u, LVP-m, LVP-e, and LVP-h all contained a large amount of bounding proteins, with 25.69%, 28.16%, 38.19%, and 28.93%, respectively, meanwhile the total carbohydrate contents in respective order were 57.59%, 55.34%, 49.95%, and 59.64%.

**Table 1.** Yields and chemical compositions of LVPs extracted by different methods.

|  | Polysaccharides Yield (%) | Total Carbohydrate Content (%) | Protein Content (%) |
|---|---|---|---|
| LVP-u | 3.74 ± 0.14 [a] | 57.59 ± 1.90 [a] | 25.69 ± 0.76 [c] |
| LVP-m | 3.09 ± 0.19 [c] | 55.34 ± 1.65 [b] | 28.16 ± 0.25 [b] |
| LVP-e | 3.33 ± 0.13 [b,c] | 49.95 ± 1.92 [c] | 38.19 ± 0.92 [a] |
| LVP-h | 3.63 ± 0.20 [a,b] | 59.64 ± 1.04 [a] | 28.93 ± 0.40 [b] |

Means within a column with different lowercase letters are significantly different ($p < 0.05$).

The molecular weight distribution of LVPs was determined by HPGPC and presented in Figure 1. The molecular weight distributions of LVPs were chromatographed and characterized as high-molecular-weight segments (Mw > 75 kDa, denoted as peak 1), medium-molecular weight segments (75 kDa > Mw > 10 kDa, denoted as peak 2), and low-molecular weight segments (Mw < 10 kDa, denoted as peak 3), based on the standard curve equation $y = 13.973 - 0.4649x$ ($R^2 = 0.9904$, $y = $ lg Mw, $x = $ retention time). As shown in Figure 1, the elution profile of each polysaccharide samples all included three peaks, as mentioned above. The area percentage ratios between the three peaks were 37.01:34.89:28.10, 30.25:38.37:31.38, 19.15:37.47:42.97, and 32.48:38.54:28.98 for LVP-u, LVP-m, LVP-e, and LVP-h, respectively.

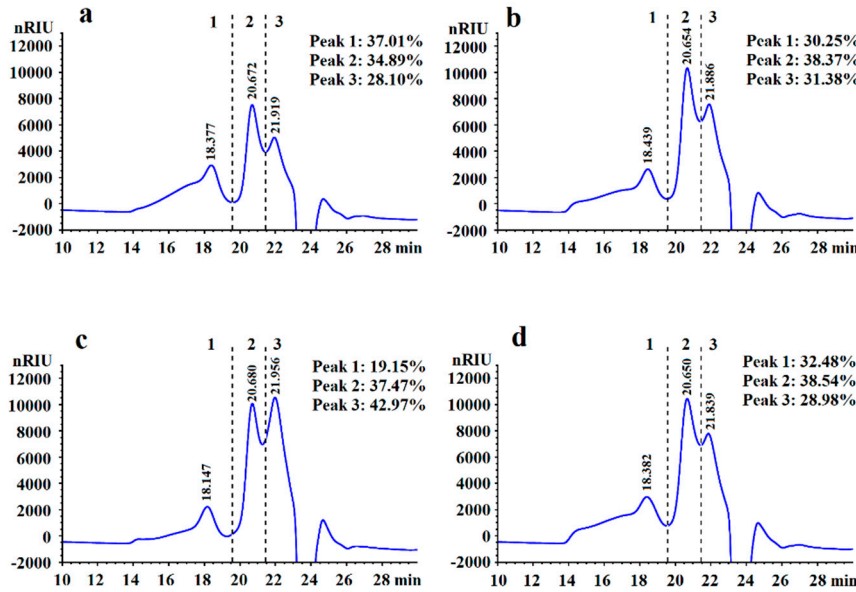

**Figure 1.** High-performance gel permeation chromatography (HPGPC) chromatogram of LVPs extracted by different methods: (**a**) LVP-u; (**b**) LVP-m; (**c**) LVP-e; (**d**) LVP-h. Elution time for different molecular weight fractions (fraction 1: Mw > 75 kDa; fraction 2: 75 kDa > Mw > 10 kDa; fraction 3: Mw < 10 kDa) are shown.

As shown in Table 2 and Figure 2, LVPs were all composed of mannose, rhamnose, glucuronic acid, glucose, galactose, and fucose. Among these, glucose and galactose were found to be the major monosaccharides constructing the backbone sugar forms in the LVPs. The molar ratio percentages of mannose, rhamnose, glucuronic acid, glucose, galactose, and fucose in LVP-u, LVP-m, LVP-e, and LVP-h were 8.41:2.60:0.75:33.01:37.23:18.00, 7.66:2.72:0.82:32.51:39.10:17.18, 11.78:3.23:1.78:24.23:43.20:15.78 and 7.71:2.60:1.32:35.09:37.71:15.56, respectively.

**Table 2.** Monosaccharides and amino acid compositions of LVPs extracted by different methods.

| | LVP-u | LVP-m | LVP-e | LVP-h |
|---|---|---|---|---|
| *Monosaccharide composition (Percentage mole ratio)* | | | | |
| Mannose | 8.41 | 7.66 | 11.78 | 7.71 |
| Rhamnose | 2.60 | 2.72 | 3.23 | 2.60 |
| Glucuronic acid | 0.75 | 0.82 | 1.78 | 1.32 |
| Glucose | 33.01 | 32.51 | 24.23 | 35.09 |
| Galactose | 37.23 | 39.10 | 43.20 | 37.71 |
| Fucose | 18.00 | 17.18 | 15.78 | 15.56 |
| *Amino acid composition (Percentage mole ratio)* | | | | |
| Asparaginic acid | 12.27 | 12.23 | 14.24 | 12.57 |
| Glutamic acid | 12.52 | 13.89 | 9.30 | 12.71 |
| Serine | 9.65 | 10.96 | 10.75 | 10.16 |
| Glutamine | 0.61 | 0.60 | 0.50 | 0.59 |
| Histidine | 1.76 | 2.50 | 1.76 | 1.85 |
| Glycine | 11.53 | 12.85 | 12.71 | 12.57 |
| Threonine | 8.44 | 9.40 | 8.95 | 8.80 |
| Arginine | 4.62 | 4.92 | 3.98 | 4.43 |
| Alanine | 11.19 | 11.75 | 10.43 | 10.72 |
| Tyrosine | 1.30 | 1.37 | 1.042 | 1.26 |
| Valine | 5.56 | 5.76 | 5.71 | 5.12 |
| Phenylalanine | 3.92 | 3.80 | 3.05 | 3.83 |
| Isoleucine | 4.39 | 4.35 | 4.07 | 3.97 |
| Leucine | 3.40 | 3.23 | 2.87 | 2.96 |
| Lysine | 4.36 | 0.71 | 4.85 | 3.77 |
| Proline | 4.46 | 1.67 | 5.79 | 4.71 |

In addition, based on the result of the BCA method, the amino acid compositions of the LVPs were also analyzed using HPLC. As shown in Table 2 and Figure 3, asparaginic acid, glutamic acid, serine, glutamine, histidine, glycine, threonine, arginine, alanine, tyrosine, valine, phenylalanine, isoleucine, leucine, lysine, and proline 16 amino acids have been detected. Their molar ratio percentages in LVP-u, LVP-m, LVP-e, and LVP-h were 12.27:12.52:9.65:0.61:1.76:11.53:8.44:4.62:11.19:1.30:5.56:3.92:4.39:3.40: 4.36:4.46, 12.23:13.89:10.96:0.60:2.50:12.85:9.40:4.92:11.75:1.37:5.76:3.80:4.35:3.23:0.71:1.67, 14.24:9.30: 10.75:0.50:1.76:12.71:8.95:3.98:10.43:1.042:5.71:3.05:4.07:2.87:4.85:5.79, and 12.57:12.71:10.16:0.59:1.85: 12.57:8.80:4.43:10.72:1.26:5.12:3.83:3.97:2.96:3.77:4.71, respectively.

The infrared spectrums of LVPs were recorded in the range of 400–4000 cm$^{-1}$ using the KBr pellet method, and the results were shown in Figure 4. The wide and strong band around at 3424 cm$^{-1}$ was attributable to the stretching vibrations of O–H and N–H, while the weak signal at 2924 cm$^{-1}$ belonged to the C–H stretching vibration [27]. A broad band at around 1640 cm$^{-1}$ corresponded to the bending vibration of an amide or amino group. The absorption peak at about 1543 cm$^{-1}$ was due to the secondary CONH group of proteins [28]. The bands at 1000–1100 cm$^{-1}$ implied the presence of pyranoside in the four samples.

Surface ultrastructure of vacuum freeze-dried LVPs was explored by the scanning electron microscopy method, and the micrographs at magnifications of 10,000× are displayed in Figure 5. The surface of LVP-h and LVP-e were smooth, whereas the surfaces of LVP-u and LVP-m were relatively rough.

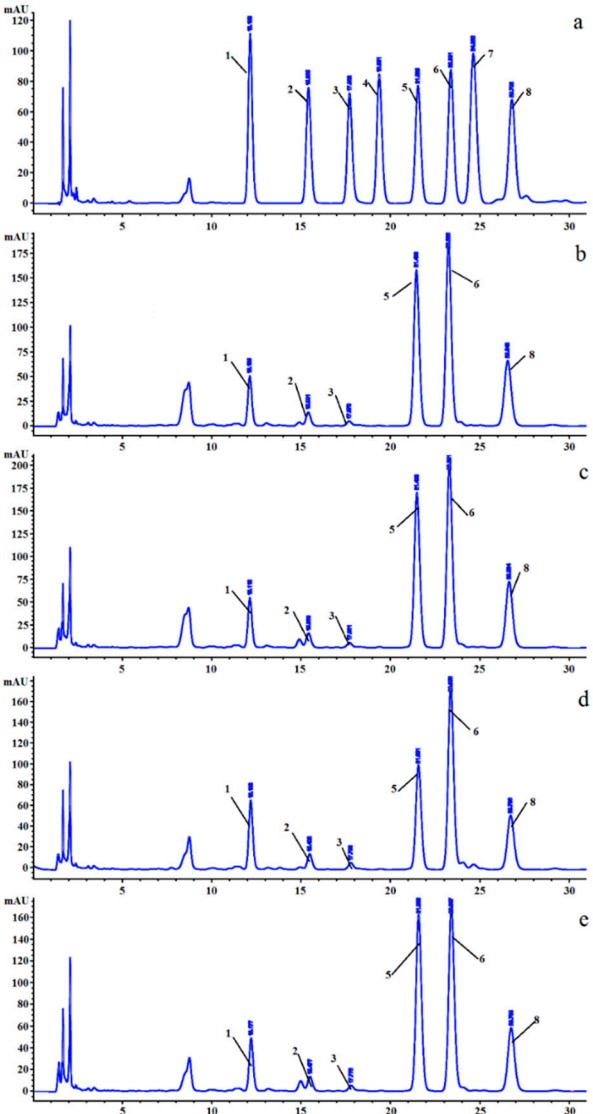

**Figure 2.** HPLC chromatograms of monosaccharide compositions of mixed standards and LVPs extracted by different methods: (**a**) mixed standards; (**b**) LVP-u; (**c**) LVP-m; (**d**) LVP-e; (**e**) LVP-h. The peaks identified are (1) mannose, (2) rhamnose, (3) galacturonic acid, (4) glucuronic acid, (5) glucose, (6) galactose, (7) arabinose, and (8) fucose.

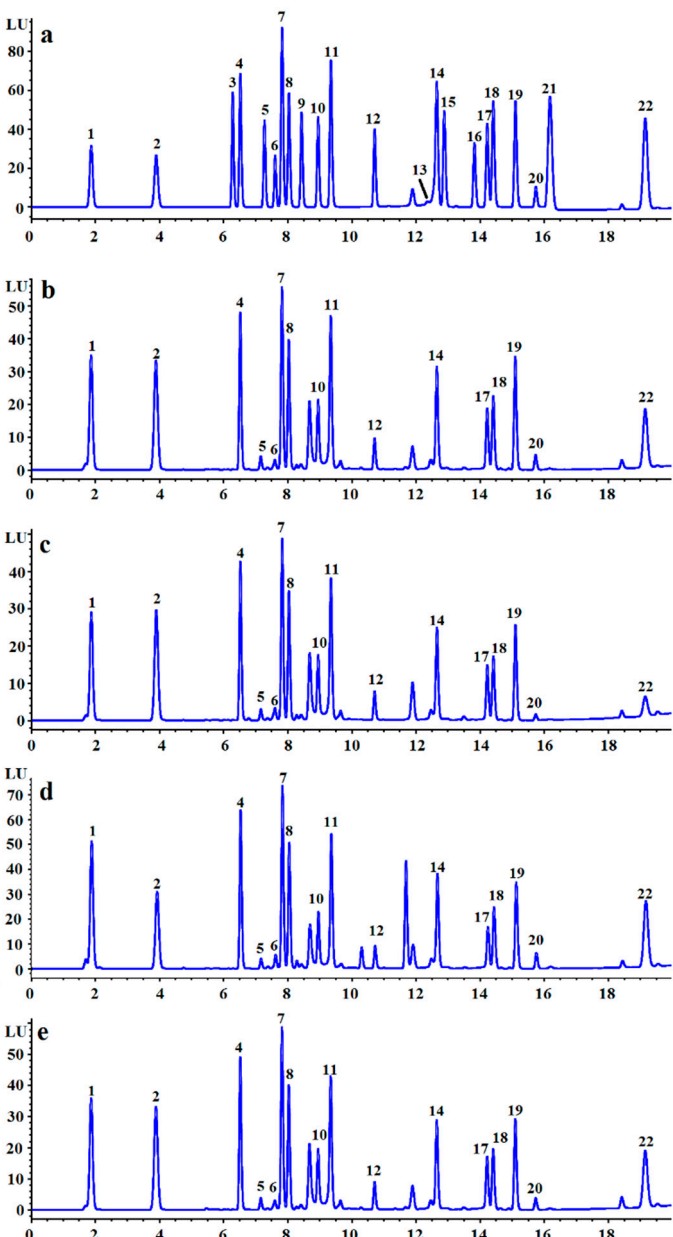

**Figure 3.** HPLC chromatograms of amino acid compositions of mixed standards and LVPs extracted by different methods: (**a**) mixed standards; (**b**) LVP-u; (**c**) LVP-m; (**d**) LVP-e; (**e**) LVP-h. The peaks identified are (1) asparaginic acid, (2) glutamic acid, (3) asparagine, (4) serine, (5) glutamine, (6) histidine, (7) glycine, (8) threonine, (9) citrulline, (10) arginine, (11) alanine, (12) tyrosine, (13) cystine, (14) valine, (15) methionine, (16) tryptophan, (17) phenylalanine, (18) isoleucine, (19) leucine, (20) lysine, (21) hydroxyproline, and (22) proline.

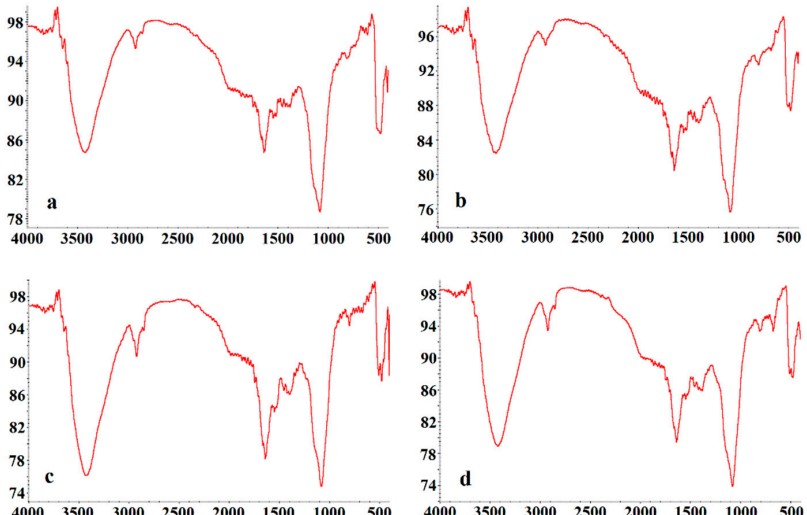

**Figure 4.** Infrared spectra of LVPs extracted by different methods. (**a**) LVP-u; (**b**) LVP-m; (**c**) LVP-e; (**d**) LVP-h.

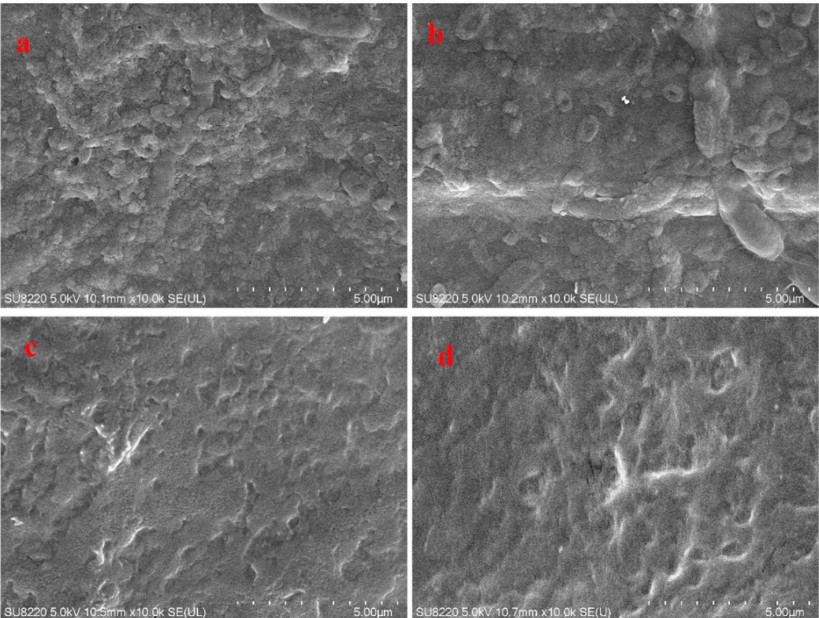

**Figure 5.** Scanning electron micrographs of LVPs extracted by different methods. (**a**) LVP-u; (**b**) LVP-m; (**c**) LVP-e; (**d**) LVP-h.

### 3.2. Radical Scavenging Activities of LVPs

In this study, superoxide anion radical and hydroxyl radical scavenging assays were conducted to evaluate the antioxidant activities of LVPs, and the experimental results were presented in Figure 6. As illustrated in Figure 6a, superoxide anion radical scavenging abilities increased with the increasing concentrations of the LVPs. At a dosage of 5.0 mg/mL, the superoxide anion radical scavenging rates of LVP-u, LVP-m, LVP-e, and LVP-h were 86.54%, 89.64%, 91.22%, and 86.63%, respectively. The median effect concentration (EC50) for the superoxide anion radicals was in the following order: LVP-u (0.83 mg/mL) > LVP-h (0.75 mg/mL) > LVP-m (0.67 mg/mL) > LVP-e (0.54 mg/mL). In addition, as presented in Figure 6b, the scavenging abilities of LVPs on hydroxyl radicals were also in concentration-dependent manners. At the maximal tested concentration (5.0 mg/mL), the hydroxyl radical scavenging rates of LVP-u, LVP-m, LVP-e, and LVP-h were 51.18%, 51.13%, 50.16%, and 52.17%, respectively, and were actually less than the Vc (99.94%) at the same treatment concentration. The EC50

for LVP-u, LVP-m, LVP-e, and LVP-h on hydroxyl radicals have been calculated as 4.95, 4.27, 4.92, and 4.05 mg/mL, respectively.

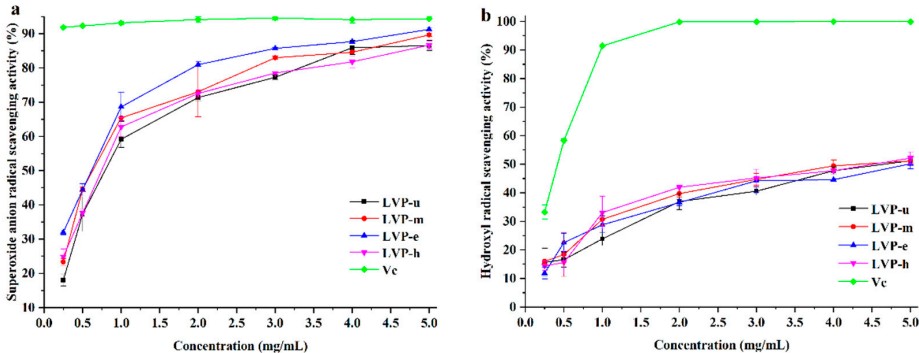

**Figure 6.** Radical scavenging activities of LVPs extracted by different methods. (**a**) Superoxide anion radical scavenging activity; (**b**) hydroxyl radical scavenging activity.

### 3.3. Metal Ion Chelating Abilities of LVPs

Ferrous ion and cupric ion chelating assays were carried out to estimate the metal ion chelating abilities of LVPs, and the results are presented in Figure 7. As shown in Figure 7a, the $Fe^{2+}$ chelating ability of LVPs increased with the increasing of LVPs concentrations, ranging from 0.25 to 5.0 mg/mL. At 5.0 mg/mL, LVP-u, LVP-m, LVP-e, and LVP-h showed a highest $Fe^{2+}$ chelating ability of 32.16%, 38.56%, 47.03%, and 47.70%, respectively. Moreover, the $Cu^{2+}$ chelating ability showed a similar trend as $Fe^{2+}$ (Figure 7b). The highest $Cu^{2+}$ chelating ability was observed in LVP-e, with 34.21% concentration at 5.0 mg/mL, followed LVP-u, LVP-m, and LVP-h, which showed 28.76%, 25.35%, and 10.41%, respectively. However, compared with the positive control EDTA-2Na, which showed percentage chelation rates of 99.42% and 96.16% for $Fe^{2+}$ and $Cu^{2+}$ at 5.0 mg/mL, the metal ions chelating abilities of LVPs were all relatively lower.

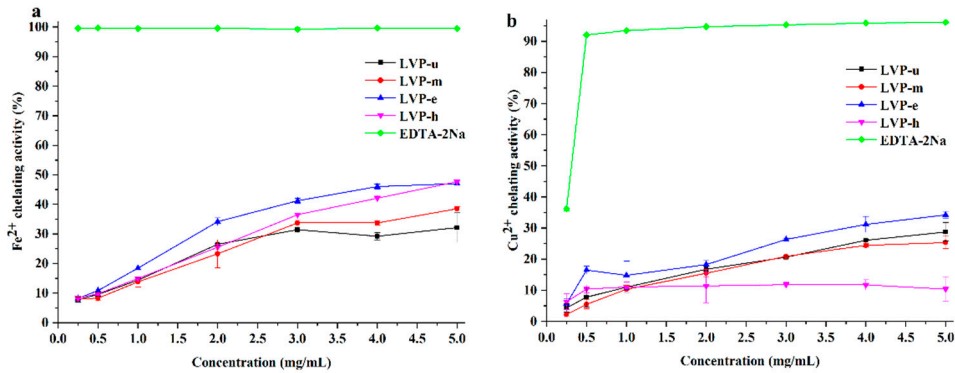

**Figure 7.** Metal ions chelating abilities of LVPs extracted by different methods. (**a**) $Fe^{2+}$ chelating ability; (**b**) $Cu^{2+}$ chelating ability.

## 4. Discussion

In eastern countries, mushrooms have been consumed as foods and medicines for thousands of years. In traditional medical theories, mushrooms such as *Ganoderma lucidum*, *Hericium erinaceus*, *Tremella fuciformis*, and *L. edodes* have been applied to clear eyesight, moisten lungs, nourish kidneys, and invigorate the spleen. Contemporarily, as a flawless nutraceutical food, mushrooms have attracted a large number of pursuers, due to their unique nutrition constituent around the world. Importantly, modern pharmaceutical chemistry research indicates that various bioactive components found in mushrooms possess the ability to regulate the immune system, lower blood pressure, and inhibit cancer, as well as other bioactive functions [3,29,30]. Among the bioactive components,

polysaccharides are significant substances that cannot be ignored. Since researchers in the 1960s found that polysaccharides purified from *L. edodes* revealed anti-tumor activity, it has been reported that polysaccharides from mushrooms have attracted wide attention for diverse biological activities (e.g., antioxidant, anticoagulant, immunomodulatory, and hyperglycemic effects) [31–34].

Extraction is the first and crucial step of polysaccharide research, and is responsible for the experimental results. Extraction methods can impact the yield, purity, chemical composition, molecular weight distribution, microstructure, and bioactivities of polysaccharides. In this study, an ultrasonic-assisted extraction method, microwave-assisted extraction method, enzyme-assisted extraction method, and hot water extraction were applied to obtain four polysaccharides (LVP-u, LVP-m, LVP-e, and LVP-h) from the fruiting bodies of *L. vividus*. Based on the experiment results, it was found that extraction methods have significant effects on the yields of LVPs, and the sequence determined was LVP-u > LVP-h > LVP-e > LVP-m (Table 1). LVP-u had the highest extraction efficiency, with a shorter time and lower energy input, which is attributed to the action of ultrasonic waves forming acoustic cavitation that generates irreversible cell wall disruption in liquid medium [35]. Meanwhile, the yield of LVP-h, with a long extraction time and high temperature, also performed well in this study. However, differences between the practical yield and expectation in the microwave-assisted extraction method have been observed, which might be due to the extracting solution absorbing microwave radiation and generating a heat effect in the extraction process, which triggers a microwave pause after the extraction temperature reaches a set point and the microwave radiation effect disappears in the meantime. Moreover, in this study, the results demonstrated that polysaccharides extracted from *L. vividus* were protein-bounding polysaccharides, and many differences were discovered among the four polysaccharides. As shown in Figure 1, the elution profiles showed three peaks for every sample, which indicates that carbohydrate fractions of the polysaccharides all contained three main components. LVP-u, LVP-m, and LVP-h showed similar percentages between the three different molecular weight segments, while LVP-e displayed the greatest difference, which indicates that polysaccharides obtained by enzyme assisted-extraction have the lower molecular weight. By monosaccharide composition analysis, the four LVPs were mainly composed of mannose, glucose, galactose, and fucose, with low rhamnose and galacturonic acid content, which makes it reasonable to conclude that LVPs are complex and heterogeneous polysaccharides. However, it can be easily found that the monosaccharide composition in LVP-u, LVP-m, and LVP-h were very close, while a difference occurred in LVP-e, for which enzyme-assisted extraction seemed prejudiced to glucose (Figure 2). These phenomena indicated that polysaccharides might be enzymolysized in the enzyme assisted-extraction process. This result is agreeable with the study that reported by Wang et al., showing that enzymes played a certain role in degradation of the polysaccharides, due to *Camellia sinensis* polysaccharides being decreased by enzyme extraction [36]. In addition, other differences, such as amino acid composition and microstructure, were also observed among the four polysaccharides in the present study (Figures 3 and 5).

In the cell, a steady flow of free radicals was generated in the process of biological metabolism, and numerous delicate balances were established to maintain the dynamic balance of these free radicals [37]. Once the source of free radicals increases or the outlet is blocked, excessive free radicals will accumulate and ultimately lead to the occurrence of oxidative stress, which can damage DNA, RNA, proteins, and lipids, resulting in an increased risk for cardiovascular disease, cancer, autism, and other diseases [38,39]. Since free radicals with unpaired electrons are highly unstable, they can directly attack biomolecules or react with metal ion catalysts, forming a hydroxyl radical which aggravates oxidative stress [40]. Thus, in vitro free radical scavenging assays and metal ion chelating assays are often utilized to evaluate the antioxidant ability of antioxidants [41]. In this study, a superoxide anion radical scavenging assay, hydroxyl radical scavenging assay, ferrous ion chelating assay, and cupric ion chelating assay were conducted to evaluate the antioxidant activities of LVPs. As shown in Figure 6, the free radical scavenging abilities varied with different concentrations in a dose-dependent manner. In general, the radical scavenging activities of polysaccharides were connected to their capacity for

anomeric hydrogen and hydrogen-donating ability [42]. In the LVPs, the abundant carboxyl and amino groups in the side chains of acidic amino acids can serve as a stable hydrogen donor. Furthermore, there are many studies demonstrating that monosaccharide composition, uronic acid content, molecular weight, and glycosidic linkage configuration of polysaccharides are also tied to the radical scavenging capacity [43–45]. Moreover, LVPs have notable metal chelating abilities for $Fe^{2+}$ and $Cu^{2+}$, which might due to the formation of a cross-bridge between divalent ions and the carboxyl group present in polysaccharides [46,47].

## 5. Conclusions

In this study, four polysaccharides from *L. vividus* were acquired by an ultrasonic-assisted extraction method, microwave-assisted extraction method, enzyme-assisted extraction, and hot water extraction. Experimental results shown that the LVPs were all protein-bounding polysaccharides, and the yields sequence was determined to be LVP-u > LVP-h > LVP-e > LVP-m. The preliminary structural properties results demonstrated that obvious variances involved in molecular weight distribution, monosaccharide and amino acid composition, and microstructure were observed. Moreover, LVP-e demonstrated better antioxidant activities in radical scavenging activity assays and metal ion chelating activity assays, while LVP-u showed higher activity in hydroxyl radical scavenging ability. Comprehensively, LVPs are natural antioxidants and potentially functional food ingredients.

**Author Contributions:** Conceptualization, C.D. and Z.X.; methodology, Z.X., S.F., and T.C.; validation and analysis, S.F., J.Q., and M.Y.; resources, C.D.; writing—original draft preparation, Z.X., J.Q., R.Y., and L.Z.; writing—review and editing, Z.X., S.F., and C.D.; supervision, C.D.

**Funding:** This research received no external funding.

**Conflicts of Interest:** The authors declare no conflict of interest.

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
