# Peer review of "The Effect of Extraction Methods on Preliminary Structural Properties and Antioxidant Activities of Polysaccharides from Lactarius vividus"

_processes, doi:10.3390/pr7080482_

Round 1
Reviewer 1 Report
Please indicate a better description of the letter referred to extraction methods, in this form is not esy to read them.
Author Response
Dear Reviewer
Thanks for your comments concerning our manuscript entitled “The effect of extraction methods on preliminary structure properties and antioxidant activities of polysaccharides from Lactarius vividus". Those comment is valuable and very helpful for revising and improving our paper. We have studied comments carefully and have made correction which we hope meet with approval.
All response parts in this letter have been written in blue font;
All revised parts in the manuscript have been written in red font.
Q. Please indicate a better description of the letter referred to extraction methods, in this form is not easy to read them.
Response: We have made some changes in the part of materials and methods.
Reviewer 2 Report
The paper is interesting for polysaccharide reseachers and mushroom eaters. However, there are four previous methods (22, 24, 25 and 26) in the Materials and Methods section and they are different authors. Please cite as Xu et al. (22); Wang et al. (24); Giese et al (25); Kalm et al. (26) in Method section. The authors Wei et al. (36) should revised to Wang et al. (36).

Author Response
Dear Reviewer
Thanks for your comments concerning our manuscript entitled “The effect of extraction methods on preliminary structure properties and antioxidant activities of polysaccharides from Lactarius vividus". Those comments are all valuable and very helpful for revising and improving our paper. We have studied comments carefully and have made correction which we hope meet with approval.
All response parts in this letter have been written in blue font;
All revised parts in the manuscript have been written in red font.
Q. The paper is interesting for polysaccharide reseachers and mushroom eaters. However, there are four previous methods (22, 24, 25 and 26) in the Materials and Methods section and they are different authors. Please cite as Xu et al. (22); Wang et al. (24); Giese et al (25); Kalm et al. (26) in Method section. The authors Wei et al. (36) should revised to Wang et al. (36).
Response: We have modified the format of the references, and revised Wei et al. (36) to Wang et al. (36).
Reviewer 3 Report
In this study, four different extraction methods (ultrasonic-assisted extraction, microwave-assisted extraction, enzyme-assisted extraction, and hot water extraction) were applied to the fruiting bodies of Lactarius vividus to obtain the polysaccharides. Preliminary structure properties, chemical composition analysis, radicals scavenging and metal ions chelating assays and antioxidant activities were analyzed.
The results demonstrated that polysaccharides extracted from L. vividus were protein-bounding polysaccharides, and many differences were discovered among the four polysaccharides related to the molecular weight distribution, monosaccharides and amino acids composition, as well as the microstructure.
Yields were different (LVP-u, LVP-h, LVP-e and LVP-m, respectively) and also the antioxidant activities (LVP-e and LVP-u) were different for each polysaccharides and, consequently, depending to the extraction method.
The paper is well organized, presented and discussed, but I would suggest the authors make corrections/modifications in their manuscript:
1. Lines 48-51: To move [10] from line 51 to after Song et al.
2. Lines 51-53: To move [11] from line 53 to after Jia et al.
3. Lines 53-56: To move [12] from line 56 to after Yan et al.
4. Line 63: HPGPC, HPLC, FT-IR and SEM: These abbreviations could be explained here. It is the first time that they are cited. Or, to cut the sentence (probably the best option): “Preliminary structural characteristics and antioxidant activities of…”
5. Line 77: o-phthalaldehyde (OPA): “o” (abbreviation of ortho-) in italics.
6. Line 80: It is weight (not weignt).
7. Line 91: “…the Sevag reagent (chloroform: butyl alcohol, 4:1, v/v) to remove…”
8. Line 103: “…(pH=5, cellulose: pectinase: trypsin the ratio of 2:2:1, v/v/v) at 50°C…”
9. Line 116: CuSO4: 4 in subscript.
10. Line 130: “The chromatographic (not detection) conditions were…”
11. Line 313: “…to obtain four polysaccharides (LVP-u, LVP-m, LVP-e and LVP-h, respectively) from the fruiting…”
12. Line 364: Please, to eliminate “respectively”.
Author Response
Dear Reviewer
Thanks for your comments concerning our manuscript entitled “The effect of extraction methods on preliminary structure properties and antioxidant activities of polysaccharides from Lactarius vividus". Those comments are all valuable and very helpful for revising and improving our paper. We have studied comments carefully and have made correction which we hope meet with approval.
All response parts in this letter have been written in blue font;
All revised parts in the manuscript have been written in red font.
Q. The paper is well organized, presented and discussed, but I would suggest the authors make corrections/modifications in their manuscript:
1. Lines 48-51: To move [10] from line 51 to after Song et al.
2. Lines 51-53: To move [11] from line 53 to after Jia et al.
3. Lines 53-56: To move [12] from line 56 to after Yan et al.
4. Line 63: HPGPC, HPLC, FT-IR and SEM: These abbreviations could be explained here. It is the first time that they are cited. Or, to cut the sentence (probably the best option): “Preliminary structural characteristics and antioxidant activities of…”
5. Line 77: o-phthalaldehyde (OPA): “o” (abbreviation of ortho-) in italics.
6. Line 80: It is weight (not weignt).
7. Line 91: “…the Sevag reagent (chloroform: butyl alcohol, 4:1, v/v) to remove…”
8. Line 103: “…(pH=5, cellulose: pectinase: trypsin the ratio of 2:2:1, v/v/v) at 50°C…”
9. Line 116: CuSO4: 4 in subscript.
10. Line 130: “The chromatographic (not detection) conditions were…”
11. Line 313: “…to obtain four polysaccharides (LVP-u, LVP-m, LVP-e and LVP-h, respectively) from the fruiting…”
12. Line 364: Please, to eliminate “respectively”.
Response: Thanks for your for your careful instruction. We have made modifications in accordance with your comments.